# Histological and Immunohistochemical Evaluation of Rh-BMP2: Effect on Gingival Healing Acceleration and Proliferation of Human Epithelial Cells

**DOI:** 10.3390/life14040459

**Published:** 2024-03-30

**Authors:** Mansour Chantiri, Samir Nammour, Sami El Toum, Toni Zeinoun

**Affiliations:** 1Department of Periodontology, Faculty of Dental Medicine, Lebanese University, Beirut 27798, Lebanon; mansour-chantiri@ul.edu.lb; 2Department of Dental Sciences, Faculty of Medicine, University of Liege, 4000 Liege, Belgium; s.namour@uliege.be; 3Department of Oral Medicine and Maxillofacial Radiology, Faculty of Dental Medicine, Lebanese University, Beirut 27798, Lebanon; samitoum@ul.edu.lb; 4Department of Oral and Maxillo-Facial Surgery, Faculty of Dental Medicine, Lebanese University, Beirut 27798, Lebanon

**Keywords:** recombinant human bone morphogenic protein 2, proliferation, epithelial cells, gingival tissue, gingival healing, Ki-67

## Abstract

This study aims to histologically and immunohistochemically evaluate the effect recombinant human bone morphogenetic protein (rh-BMP2) injected in gingival tissue has on the acceleration of the epithelial migration from the wound edges and epithelial cell proliferation after implant surgery. Material and Methods: The study includes 20 patients who underwent bilateral implant surgeries in the premolar-molar region of the mandible, followed by guided bone regeneration. Each patient received an implant in both locations, but rh-BMP2 was only on the right side. At 9 days from the surgery, a gingival biopsy was performed 3 mm distally to the last implant. In total, 20 samples were collected from the left side (control group #1) and 20 from right (test group #1). This was repeated at a 4-month interval during healing abutment placements. Tissues were processed and stained with hematoxylin-eosin and then immunohistochemically for the expression of Ki-67 and further histological examination. Result: Complete closure of the epithelium with new cell formation was observed in the 55% test group and 20% control group after 9 days. At 4 months, although 100% samples of all groups had complete epithelial closure, the test group showed that the epithelial cells were more organized and mature due to the increased number of blood vessels. The average number of new epithelial cells was 17.15 ± 7.545 and 16.12 ± 7.683 cells per mm in test group, respectively, at 9 days and 4 months and 10.99 ± 5.660 and 10.95 ± 5.768 in control groups. Conclusion: Evident from histological observations, rh-BMP-2 can accelerate the closure of gingival wounds, the healing process of epithelial gingival tissue, and the formation of epithelial cells in patients undergoing dental implant treatment.

## 1. Introduction

A group of 20 proteins linked to transforming growth factors are known as bone morphogenic proteins (BMP). They differentiate osteoblasts from mesenchymal cells and help perform many cellular activities like adhesion, differentiation, migration, and proliferation in addition to programmed cell death [1]. BMPs can produce bone after binding to receptors found on various cell types of plasma membranes (autocrine and paracrine effects), establishing cell and tissue organization [2].

BMPs have an essential role to play in bone formation at the time of fracture repair. Members of the superfamily transforming growth factor-β, at least 15 BMP, have been identified up to the current date. BMP2 (most commonly used in dentistry), BMP3 (osteogenic), BMP4, and BMP7 (osteogenic protein-1) are frequently studied. A recombinant type of BMP has already been synthesized (rh-BMP) [3].

BMPs involved in the regulation of various cellular processes including embryonic development tissue differentiation and bone formation. BMPs are a subclass of the transforming factor-beta superfamily. They play essential roles in embryogenesis, organogenesis, and tissue homeostasis. Dysregulation of BMP signaling has been implicated in various diseases, including cancer and skeletal disorders.

The rh-BMP2 can be used in the Oro-facial area in maxillary sinus augmentation [4], in alveolar ridge augmentation and preservation [5,6], in the alveolar defect in cleft palate [7], and in implant and peri-implantitis [8,9].

Smith et al. [10] confirmed that the formation of new epithelial cells in the gingiva is crucial for wound healing. Epithelial cells play a key role in the regeneration of gingival tissue, forming a protective barrier over the wound. This process is essential for proper healing and restoring the integrity of the gingiva after injury or surgery.

BMPs stimulate bone and cartilage formation (BMP2) and initiate endochondral bone formation (BMP3, BMP7) [11]. Also, BMP3 plays a role in the development of the central nervous system. BMP2 and BMP4 in RNA transcripts have been found to play a role in developing tooth buds, the odontoblast layer, and other craniofacial structures. BMPs are also involved in the extracellular matrix during repair and regeneration; they stimulate angiogenesis by extracellular matrix interactions [12,13,14,15,16,17].

Gerdes et al. [18] originally identified the Ki-67 antigen using a monoclonal antibody, generated by immunizing mice with the nuclei of Hodgkin’s lymphoma cell line. Detailed analyses showed that the Ki-67 antigen was present in the nuclei of cells during all phases of the cell cycle, whereas it was not expressed in quiescent or resting cells in the G0 phase. Mang et al. [13] considered this protein as an excellent marker for determining the growth fraction of a given cell population due to its absence in quiescent cells and universal expression in all proliferating cells (normal as well as tumor).

At the gingival level, several variables can be used to evaluate the impact of rh-BMP-2, e.g., the vascular endothelial growth factors (VEGF) in the epithelium and connective tissue graft, as well as the Ki-67 protein, which has been widely used as a proliferation marker [19,20,21,22,23] Antibodies against Ki-67 have been used as prognostic factors in the diagnosis of several types of neoplasm [18,23].

According to Smith et al. [24], gingival wound healing includes a chain of sequential responses and steps that help in the closure of breaches in the masticatory mucosa. This is essentially important to stop the microbes from entering tissues, and thus it prevents a chronic infection. The epithelium surface appears to be healed after 10 days. Wound healing may also have a critical part to play during cell and tissue reactions in the event of a long-term injury, as it may occur during inflammatory responses and cancer. Furthermore, some studies demonstrate that gingival wound healing is adversely affected by the aging process. Such defects change individual phases of the wound-healing process, like epithelial migration, granulation tissue formation, as well as tissue remodeling. The cellular and molecular defects that show these deficiencies are as follows: an inflammatory response more than normal, change in integrin signaling, diminished activity of growth factor, cell proliferation less than normal, reduced angiogenesis, decreased collagen synthesis, augmented collagen remodeling, and weakening of the proliferative and differentiation potential of stem cells. Hence, it can be concluded that this process is essential for proper healing and restoring the integrity of the gingiva after injury or surgery.

The literature review shows that most of the research focused on the effect of rh-BMP-2 on bone healing, but not on the acceleration of the wound edge and its closure. A better and faster gingival cell proliferation leads to better gingival healing, reduces infection, and provides a good environment for favorable wound healing.

This study aims to investigate the effect of rh-BMP-2 injection in the gingiva and bone grafting material on the acceleration of the gingival healing process and the proliferation and formation of new epithelial cells in the gingival tissue. The null hypothesis is that rh-BMP2 has no effect on the acceleration of gingival healing and on the proliferation and formation of new epithelial cells in the gingival tissue.

## 2. Materials and Methods

This study protocol has been approved by the ethical committee of the Lebanese University under the number CUER 27-2020 of Hadat, 20 of August 2020, and the randomized control trial registration number: LU-DP-5.2.20 of 21 September 2020. All patients were initially interviewed and examined for eligibility after obtaining written informed consent.

### 2.1. Study Design

This is a clinical and histological research. This randomized control clinical trial, performed in Lebanese University included 20 patients (20) selected consecutively following the inclusion and exclusion guidelines. Each patient received one or two implants in the left and right molar and premolar mandibular regions followed by GBR

“GBR was performed regarding the bone dehiscence and exposing of the implant exposing buccally or lingually”.

On the right side, an injection of 25 µg of rh-BMP2 during implant surgery and GBR in the gingival tissue was performed, and on the left there was no injection. On the 9th day, during the suture removal, a biopsy was performed 3 mm distally to the distal implant on the left side (control group # 1) and the right side (test group #1). A total of 40 biopsy samples were collected, 20 from each side. After 4 months of follow-up during the second stage of implant, the punched tissue collected was used as a biopsy specimen. The left side was considered as the control group #2 and the right side as the test group #2.

The specimens were stained with hematoxylin and immunohistochemistry for the Ki-67 marker. Expressions of the marker Ki-67 were counted in cells in each group. The value of ki-67 expression is calculated by counting the cells colored in purple and in brown and represents the ratio of new cells and total cells. A high Ki-67 expression value has a larger number of new cell formations and is, therefore, likely to grow faster.

### 2.2. Study Population

In total, 20 healthy patients (n = 20), with an age more than 21, gender notwithstanding, were selected from those consulting the Faculty of Dental Medicine (FDM) at the Lebanese University for a chief complaint to replace premolar and molar mandibular missing teeth (bilateral posterior edentulous mandibular).

Patients applied at the FDM starting with a medical observation, mucosal, and dental clinical examination at the Oral Diagnostic Service, and then the requested X-rays were performed at the Oral Radiology Service. Based on the clinical and radiological data, the patients received the proposed treatment plan, which was either placement of dental implants replacing missing posterior teeth or a conventional fixed or removable prosthesis. Patients decide the treatment plan according to their best convenience after receiving a detailed explanation about the different advantages and disadvantages of each option. Patients who decided to undergo the dental implant treatment option were referred to the Periodontology Department at FDM and were re-examined to decide the appropriate technique to ensure the success of the therapy. These patients had implants placed on both sides of the mandible and were given an injection of rh-BMP2 only on the right side. All subjects who failed to show up but who provided informed consent to participate in this study were reported regardless of whether implant placement had taken place. All subjects were scheduled to participate in the investigation in consecutive order, (Figure 1) provided they fulfilled the criteria stated in this protocol.

### 2.3. Inclusion and Exclusion Guidelines

Patients were selected according to the following guidelines:

Inclusion guidelines

-Men or women more than 21 years of age.-Not leaving the city for at least coming 4 months.-Ability to give written informed consent by self.-Either light smokers (not more than five cigarettes per day) or without the habit.-Patient with bilateral edentulous mandibular posterior ridge presenting with at least buccolingual bone width of 5 to 6 mm, calibrated with Cone Beam Computed Tomography (CBCT).-Bone graft required in the molar and premolar areas on both mandible sides during implant placement.

Exclusion guidelines

-Patients vulnerable to infections: positive for one or more known infectious diseases like Human Immunodeficiency Virus (HIV), hepatitis, and infectious mononucleosis.-Suffering from a clinically severe systemic disease.-Diabetic.-Patients at risk of endocarditis.-Patients using anti-thrombotic medication like heparin, and anti-vitamin K.-Poor understanding of any of these languages, written or spoken: Arabic/English/French.-Diseases that involve frequent intake of antibiotics.-Patients with proven hypersensitivity to BMPs.-Patients who are already participating in another interventional study.-Patients with allergy/hypersensitivity to both amoxicillin, clindamycin, and local anesthetics.-Pregnancy or planning for it.-Lactating mothers.-Excessive smokers

### 2.4. Implant Placement Procedure

An intra-sulcular incision with a 15 C blade, starting from the mesial part of the anterior tooth and continuing backward at the mid-cretal level of the edentulous region until the second mandibular molar was performed. An elevation of full-thickness mucoperiosteal flap was performed and a sequence of drills was used as recommended by the manufactory to permit the placement of an INNO implant of 4.0 mm in diameter and 8 to 10 mm in length. The implant was placed with an implant holder and handpiece torque (torque 30 N/cm).

The placement of an implant of 4 mm in a 6 mm or less ridge width led to a perforation of the cortical buccally or lingually and bone dehiscence. A GBR was recommended.

### 2.5. GBR

A bovine bone graft (Dia Bone, Cowell Medi Co., Ltd., Seoul, Republic of Korea) was placed regarding the implant’s exposed surface and covered with a cytoplast membrane (Dia membrane Cowell Medi Co., Ltd., Seoul, Republic of Korea). The flap was closed with separated horizontal sutures.

On the right mandibular side, 1 mL of soluble rh-BMP2 in a physiological serum with a concentration of 0.25 µg/mL (Cowell Medi Co., Reference: BB 1025, Seol, Republic of Korea) was injected into the bone graft material and gingival tissue by using a syringe of 21 G needle (Weigao Medical International Co., Ltd., Suzhou, China).

An identical protocol of guided bone grafting was performed on the left mandibular side without injection of rh-BMP2. Both sides were treated simultaneously in one session.

### 2.6. Biopsy Procedure

At 9 days after the surgery during suture removal, a biopsy was performed. An injection of 0.5 mL of analgesia Articaine 4% (Septanest with adrenaline 1/100,000, Saint Bernard, United Kingdon) was injected distally to the implant.

-The distal part of the implant was localized with a periodontal probe.-The biopsy was performed using the biopsy forceps from the gingiva 3 mm distally from the implant at the incision level. The oval collected biopsy dimension was 2–3 mm of diameter and around 2 mm of thickness.-Fixation of the specimen in 10% buffered formalin was performed and the sample was sent to the laboratory.

At 4 months after the surgery, and during the second implant stage, a biopsy was performed:-Injection of 0.5 mL of analgesia Articaine 4% (Septanest with adrenaline 1/100,000, Saint Bernard, United Kingdon) buccally and lingually to the last implant.-Localization of the distal of the implant with a periodontal probe.-Excision of the tissue covered the head of the implant using a tissue puncher of 4 mm diameter (Cowell Medi Co, Ltd., Seoul, Republic of Korea). The biopsy was circular with a 4 mm of diameter and a thickness of around 2 mm.-Fixation of the gingival specimen in 10% buffered formalin was performed and the sample was sent to the laboratory.

### 2.7. Histological and Immunohistochemical Procedures

Tissues were fixed in 10% buffered formalin and routinely processed and embedded with paraffin. Two or three serial sections of 4 μm thickness were prepared and put upon salinized slides. Xylene and descending grades of alcohol were used to deparaffined and rehydrate the sections. Beginning from 2–5 min, antigen retrieval was performed using a pressure cooker in 10 mM citrate buffer (pH 6.0). To control any endogenous peroxidase activity, the sections were covered with 3% hydrogen peroxide and incubated for about 15 min. Then, the twenty slides were incubated for 4 h at room temperature with primary anti-Ki-67 Rabbit polyclonal antibody (Abcam Inc., Cambridge, MA, USA) at a dilution of 6 μg/mL After a sequence of incubating first for 45 min with the secondary antibody followed for 30 min with streptavidin peroxidase and then for 10 min with freshly prepared diaminobenzidine (DAB) chromogen to permit the visualization. Examination and counting of the staining slides with Harris hematoxylin were performed under a microscope with 40× magnification to calculate the Ki-67 expression value.

As per the National Center for Biotechnology Information at U.S. Information (NCBI), the purple color of the cells indicated new cell forming and the brown color of old cells. The Ki 67 expression value of the whole slide was calculated using the mean value of counting the number of stained cells in five fields. In total, 200 slides (20 × 5 × 2 = 200) were examined on day 9 and the same number of slides after 4 months.

### 2.8. Evaluation of Healing

#### 2.8.1. Healing

The evaluation of the healing was performed through histological examination of the slide under the microscope magnification of 40× to explore incomplete or complete closure of the epithelium surface and cell organization and maturation at 9 days and 4 months, respectively.

#### 2.8.2. Evaluation of Epithelial Cell Number

New cells were considered positive for ki-67 antibody if there was purple intracellular staining irrespective of its intensity, while cells without purple staining were considered negative cells (old cells) Five of the most invading islands of the biopsies were captured and the total number of the newly formed cells were counted vertically from one side to another. The labeling index for each field (field LI), was calculated as the total number of positive cells divided by the total number of cells in the field and for each specimen, the mean value of the five islands was calculated to represent the mean expression value of each specimen. For each group in this research project, a mean value of the labeling index was calculated.

### 2.9. Statistical Analysis

For statistical analyses Prism 9.5.1. 733^®^ software (GraphPad Software, Inc., San Diego, CA, USA) was used. It was considered statistically significant when the *p*-value was less than 0.0001 and highly significant when the confidence level was proposed to be 99% with *p* > 0.001. The calculation of mean and standard deviation values of the new epithelial cells were performed for control and test groups at 9 days and at 4 months. The normal distribution of the variables was performed with 4 normality tests (Agostino, Pearson, Anderson-Darling, Shapiro–Wilk, and Kolmogorov–Smirnov tests). In non-normal distributions, Friedman tests were used for non-parametric and repetitive measurements coupled with Dunn’s multiple comparisons test (ad hoc test). Pearson’s Chi-Square Test was used to assess the relationship of rh-BMP2 injection and wound closure.

## 3. Result

Ki-67-stained purple cells indicated they are in the replicating or dividing process (new cells formation) and brown-stained are old cells. Cells-stained purple indicates a higher level of cell proliferation compared to cells-stained brown, which indicates a lower level of cell proliferation.

### 3.1. Epithelial Healing

The evaluation of the epithelium healing histologically showed 20% of biopsies (4 biopsies) with completely closed epithelium surfaces in the control group #1 and 55% (11 biopsies) in the test group #1 (Table 1). Incompletely closed epithelium was 80% (16 biopsies) in the control group #1 and 45% (9 biopsies) in the test group #1, as shown in Table 1. The distribution of complete and incomplete healing in each group is illustrated in Table 1. Since the calculated Chi-Square statistic (5.2267) exceeds the critical value of 5.024 at 0.025 significance level, in the Chi-square distribution table, the null hypothesis (H0) is rejected. The acceleration of wound closure at 9 days intervals differs significantly between test and control groups #1, suggesting a potential effect of the rh-BMP-2 product on the epithelial healing process and acceleration of the wound closure.

At 4 months, the evaluation of the epithelium healing under the microscope of magnification × 40 showed 20 biopsies (100%) with complete closed epithelium surface in the control group #2 (Figure 2) and in the test group #2. Figure 2 shows that the epithelial cell maturation and organization were much improved in test group #2 than in control group #2.

### 3.2. Ki-67 Expression and Number of New Epithelial Cells

The mean value of Ki-67 expression value in the control groups was 10.99 ± 5.66 cells per mm^2^ on the 9 days and 10.95 ± 5.768 cells per mm^2^ at 4 months. While the test group was 17.15 ± 7.545 cells per mm^2^ at the 9 days and decreased to 16.12 ± 7.683 cells per mm^2^ (Table 2, Figure 3).

The statistical analyses showed a significant difference among the test and control groups at 9 days and 4 months (Table 2 and Figure 3).

The results of this study showed that the wound closure and the new epithelial cell formation in every group were significantly higher in the test groups where rh-BMP-2 is used than in the control groups. For this reason, the null hypothesis was rejected.

## 4. Discussion

This research showed that BMPs play a crucial role in promoting gingival cell growth and tissue regeneration. They have been shown to enhance faster epithelium closure and the proliferation of epithelial gingival cells contributing to gingival tissue repair and better wound healing in the gingiva.

It is well known that the formation of new epithelial cells results in faster healing for the gingival tissue [10]. This is why the number of new epithelial cells formed was used in this study to evaluate the healing process of the gingiva. Increasing the number of new epithelial cells enhanced wound healing. The rh-BMP-2 injection epithelial surface closed completely on the 9 days in most cases (55%) and at 4 months after surgery, ensuring better quality of the gingiva which in turn provides excellent prosthetic outcomes [10]. A complete healing of the epithelial surface at 9 days confirms faster healing.

Scheyer et al. [24] reported that adverse repercussions of using BMP2 in oral/dental surgery and GBR treatment were scarce in the literature, probably because the long-term complications were not included as a variable in conducted clinical studies and the quantity used was significantly lower than the one used in other general surgeries and therapeutic procedures.

Park et al. [25] reported that in contrast to protein delivery, BMP2 gene transfer into the defect site induces BMP2 synthesis in vivo and leads to secretion for weeks to months, depending on the vector, at a concentration of nanograms per milliliter. BMP2 gene delivery is advantageous for the bone wound healing process in terms of dosage and duration. However, safety concerns related to viral vectors are one of the hurdles that need to be overcome for gene delivery to be used in clinical practice.

Few studies reported complications after using BMP2, chiefly in spine surgeries and bone repair, and related to the concentration and dosage of BMP2 used [26,27].

Chantiri et al. [28] demonstrated that the injection of rh-BMP-2 in the gingiva induced a significant increase in newly formed blood vessels ensuring prompt healing.

A strong correlation between the intensity of proliferating cells and the rate of gingival healing was also discovered. Higher rates of proliferation and new cells formation are associated with faster rates of healing [29,30,31,32].

The staining of Ki-67 on gingival tissue is typically assessed by evaluating the percentage of cells that are stained in the tissue sample. The staining of Ki-67 is usually detected using immunohistochemistry (IHC), and the cells appear brown or purple in color. This study included 20 samples collected from a mandibular side injected with rh-BMP-2 during the implant surgery and 20 from the contralateral side which did not receive any injection. The sample collection was performed at 9 days and after 4 months, and it showed an increase of the purple stained cells in the injected side.

Rh-BMP2 is a growth factor that plays an important role in the regulation of bone and tissue formation. It has been used clinically to stimulate bone formation in various applications, including spinal fusion, fracture healing, and dental bone augmentation. outcomes [28,29,30,31].

Recently, rh-BMP-2 has been studied for its effects on the proliferation of gingival cells. Research has shown that rh-BMP-2 promotes the proliferation of gingival cells, which can be beneficial for a variety of dental treatments. It has also been demonstrated that rhBMP-2 can increase the proliferation of osteoblasts, which has possible applications for dental bone augmentation.

Several studies have shown that rh-BMP-2 can stimulate gingiva cell proliferation in vitro in a laboratory setting. The effects of rh-BMP-2 on gingival cells have also been studied in vivo, in living organisms, with promising results [28,32].

Another study demonstrated that rh-BMP-2 promoted gingival tissue regeneration in a rabbit model, with increased proliferation and matrix production [33].

Overall, rh-BMP-2 is an effective growth factor for promoting gingival cell proliferation in both in vitro and in vivo settings. This suggests that rh-BMP-2 may be a useful tool in a variety of dental procedures [28,32,33].

To increase the osteogenic activity of implant surfaces. Behrens et al. [30] developed a collagen/heparin-based multilayer coating on titanium surfaces. This coating delays the release of rh-BMP-2. They concluded that the Nano coating using collagen/heparin-based PEMs can incorporate rh-BMP-2 in significant levels upon titanium surfaces with delayed release and a sustained enhancement of osteogenic activity without altering the surface morphology.

Kawecki et al. [31] reported that a BMP supplementation could enhance the in vitro osteogenic potential of the bone-like substitute, which later helps in better repair of alveolar bone repair after extraction.

Micro-computed tomography and histological analyses sometimes demonstrate the same, or even better, global alveolar bone preservation when defects were filled with BMP-9-treated bone-like substitutes for a duration of about 10 weeks in contrast to a clinical-grade biomaterial, with adequate gingival re-epithelialization in the absence of resorption [29].

Cell-matrix interactions are essential for epithelial cell formation. In non-wounded tissue, basal epithelial cells interact with multiple parts of the intact basement membrane. After surgery, keratinocytes are exposed to multiple arrays of matrix components which are type I collagen, polymerized fibrin, and plasma fibronectin [30].

A future study on the effect of BMP on gingival tissue proliferation could investigate its impact on cell growth, tissue regeneration, and potential applications in periodontal health. Future studies should consider designing experiments to analyze cell cultures, animal models, or clinical trials to assess BMP’s role in promoting gingival tissue proliferation and its relevance to dental therapies.

Therefore, several biological responses deserve further and multiple studies to characterize the wound-healing defects associated with rh-BMP2.

## 5. Conclusions

The results of this study showed that the gingival injection of rh-BMP2 accelerates gingival tissue healing and the proliferation of new epithelial cells in patients undergoing dental implant treatment.

## Figures and Tables

**Figure 1 life-14-00459-f001:**
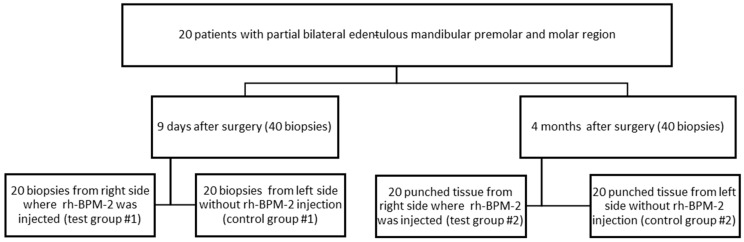
Study Design.

**Figure 2 life-14-00459-f002:**
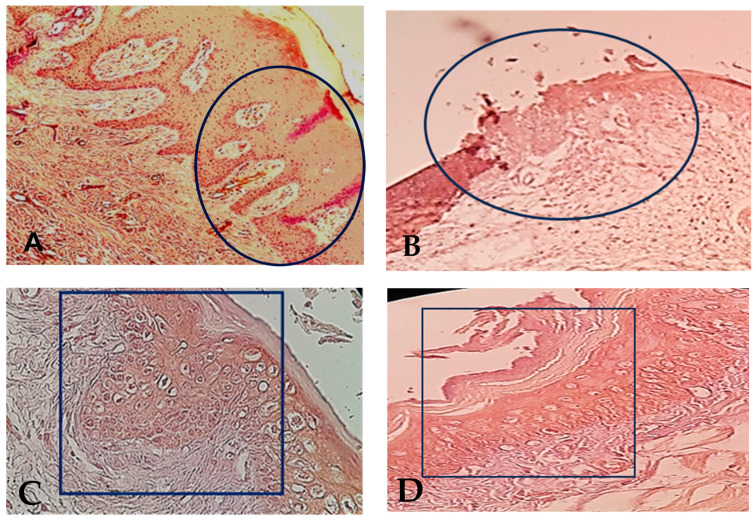
(**A**): Complete epithelial healing in test group #1, 9 days after surgery. The black circle indicates the complete epithelial closure at the biopsy site. The suture area has been framed with regular organization of the epithelial cells at magnification × 40. (**B**): Incomplete epithelial healing in the control group #1, 9 days after surgery. The black circle indicates the non-closed epithelial layer with irregular organization of the epithelial cell layers, at magnification × 40. (**C**): The black rectangle indicates the complete closure of the epithelium surface in test group #2 at 4 months after surgery with improvement of cell maturation of the epithelial gingival tissue after total healing (magnification × 40). (**D**): The black rectangle indicates the complete closure of the epithelium surface. in control group #2 at 4 months after surgery with normal epithelial gingival tissue organization (magnification × 40).

**Figure 3 life-14-00459-f003:**
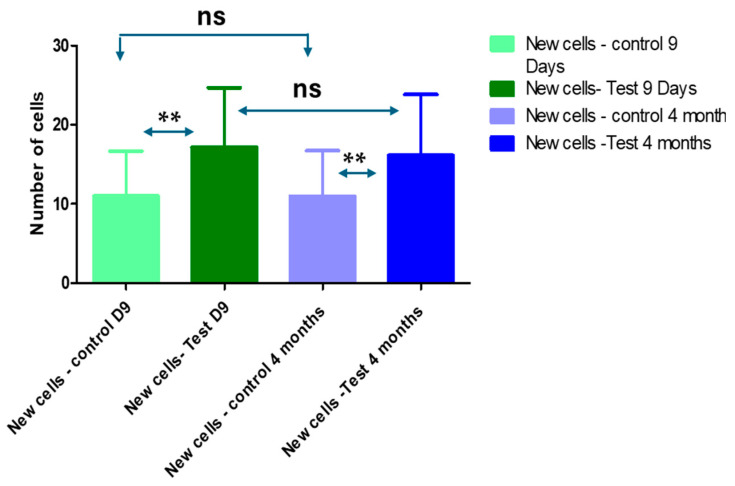
Average of new epithelial cells in the control and test groups at 9 days and 4 months. Friedman tests for non-parametric and repetitive measurements coupled to Dunn’s multiple comparisons test (ad hoc test) showed significant differences among the control and test groups at 9 days and 4 months (ns: non-significant, **: highly significant).

**Table 1 life-14-00459-t001:** Number and percentage of wound closure in every group. Pearson Chi-Square test revealed a statistically significant relationship between acceleration of wound closure and BMP2 injection (*p* = 0.022).

	Number of Wounds in the Group without Injection of rh-BMP2 (Control Group #1)	Number of Wounds in the Group with the Injection of rh-BMP2 (Test Group #1)
Incomplete	16	9
Complete	4	11
Percentage of complete wound closure	20%	55%

**Table 2 life-14-00459-t002:** The mean and standard deviation (Std) of the number of new epithelial cells formed in each group are shown. Non-identical superscript letters (A and B) express a statistically significant difference (*p* Value < 0.0001) while the identical superscript letters express the absence of a statistically significant difference.

	Day 9 after Surgery		4 Months after Surgery	
New Epithelial Cells	Control Group # 1 without rh-BMP-2	Test Group #1 with rh-BMP-2	Control Group # 2 without rh-BMP-2	Test Group #2 with rh-BMP-2
Mean	10.99 ^A^	17.15 ^B^	10.95 ^A^	16.12 ^B^
Std. Deviation	5.660	7.545	5.768	7.683

## Data Availability

The data can be shared upon request.

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
