# Peer review of "Histological and Immunohistochemical Evaluation of Rh-BMP2: Effect on Gingival Healing Acceleration and Proliferation of Human Epithelial Cells"

_life, 2024, doi:10.3390/life14040459_

Round 1

Reviewer 1 Report

Comments and Suggestions for Authors

Firstly, I would like to congratulate you on your work. You have shown with a population of 20 patients that rh-BMP-2 injection can improve wound healing. Overall, however, your study seems a little disorganised to me, so that some improvements are still needed before publication in Life.

Major issues:

You write that the biopsy was taken AFTER the suture removal. Where exactly was the biopsy taken and how was this ethically justifiable?

At what point after 4 months was the second biopsy taken "during the second stage of the implant"?

It would be nice if you could show an illustration of the surgical procedure or an exemplary photograph of the operation.

You write briefly about the BMP superfamily (L44-48) in the introduction, but you should explain this in more detail.

The introduction contains many aspects that belong more into Discussion. For example, you mention other authors who use BMP2 - and complications (L65-78).

The discussion should begin with a brief summary of the main findings

Minor issues:

L. 130: upper/ lower case letters

Figure 2, 3, 5: Please add a scale bar

Figure 4: You should use either numerical or percentage data

Figure 7:  Brackets with asterisks should be used to indicate the significance.

L. 315: mm²

Author Response

Dear Reviewer,

Reviewer 1

I hope that your recommendation will help to improve the quality of our manuscript. Thank you

  • You write that the biopsy was taken after the suture removal. Where exactly was the biopsy taken and how was this ethically justifiable?

Response:- The biopsy was taken when suture removal at “9 days” because during the surgery, we performed a thick gingiva distally to the last implant, and regularization of this gingiva permitted us to get the sample required for the study. The biopsy was done 3 mm distally of the  implant at the incision level. Our patients had been informed about the procedures and they signed the consent form.

However, the following descriptive sentence was added in the manuscript:

The biopsy was done using the biopsy forceps from the gingiva 3 mm distally from the implant at the incision level. The tissue collected is 2-3 mm of diameter. “

  • At what point after 4 months was the second biopsy taken "during the second stage of the implant"?

Response: -The second biopsy after “4 months” during the second stage surgery of the implant,. was collected with a tissue puncher on the head of the Implants.

The following sentence was added in the manuscript:

“ Excision of the tissue covered the head of the implant using a tissue puncher of 4 mm diameter.”

3- It would be nice if you could show an illustration of the surgical procedure or an exemplary photograph of the operation.

Response : we didn’t add the photo of surgery in the manuscript because our target was the histological issue and not clinical and because we had a lot of illustrations.

For your info, we show here below some photos of our procedure.

.

 (A)         (B)                                                                

 (C)         ( D)

 E

(E) Infiltration of rh-BMP2 after implant surgery at day 0...

(F)  (G)

Full Surgical Procedures from implant placement to GBR with 2nd stage

(A)implant bed preparation with bone dehiscence (B-C-D)implant placement, GBR and suture  (E) BMP2 infiltration after implant surgery..

 (F -G) After 4 months: Biopsy by punch to discover the implant at the second stage  and the placement  of the abutment.

4-You write briefly about the BMP superfamily (L44-48) in the introduction, but you should explain this in more detail

Response:.  Following Your request, we added the following paragraph in the manuscript:L54-59.

“BMP superfamily is a group of proteins involved in the regulation of various cellular processes including embryonic development tissue differentiation and bone formation. BMP’s are a subclass of the transforming factor-beta superfamily. They play essential roles in embryogenesis, organogenesis, and tissue homeostasis. Dysregulation of BMP signaling has been implicated in various diseases, including cancer and skeletal disorders.”

5- The introduction contains many aspects that belong more into Discussion. For example, you mention other authors who use BMP2 - and complications (L65-78).

Response. The paragraph of the introduction L65-78, L390-404.  was moved to the discussion.

Scheyer et al. [25] reported that adverse repercussions of using BMP2 in oral/dental surgery and GBR treatment were scarce in the literature probably because the long-term complications were not included as a variable in conducted clinical studies and the quantity used was significantly lower than the one used in other general surgeries and therapeutic procedures.

Park et al [26] reported that in contrast to protein delivery, BMP2 gene transfer into the defect site induces BMP2 synthesis in vivo and leads to secretion for weeks to months, depending on the vector, at a concentration of nanograms per milliliter. BMP2 gene delivery is advantageous for the bone wound healing process in terms of dosage and duration. However, safety concerns related to viral vectors are one of the hurdles that need to be overcome for gene delivery to be used in clinical practice.

Few studies reported complications after using BMP2, chiefly in spine surgeries and bone repair, and related to the concentration and dosage of BMP2 used ]27,28].

 6- The discussion should begin with a brief summary of the main findings.

Response:: a brief summary of the main findings was added at the beginning of the discussion as follows. From L377-380:

“The result of this research suggests that BMPs play a crucial role in promoting gingival cell growth and tissue regeneration. They have been shown to enhance faster epithelium closure and  the proliferation of epithelial gingival cells contribute ng to gingival tissue repair and better wound healing in the gingiva.’

Minor issues:                                           

  1. 130: upper/ lower case letters,

Response  : Figure 1: Study Design. L 148-149.

  •  
  • Figure 2, 3, 5: Please add a scale bar

Response:  on the histological figures, we didn’t add the scale bar because we are not searching for any measure. Our target is to evaluate the complete closing of the epithelium and the counting of the number of cells.

  • Figure 4: You should use either numerical or percentage data.

Response: We totally agree with you. But, following another reviewer recommendations, we removed the figure 4 because the table 2 is clear enough.

  • Figure 7:  Brackets with asterisks should be used to indicate the significance.

Response: we agree with you. We erased the table 3 and added the significance on the figure 7.

  • Table 3:  Brackets with asterisks should be used to indicate the significance.

Response: we agree with you. we deleted the table 3 and added the significance in the legend of  the new Figure 3.

  • 315: mm².

    Response L 351: per mm 2

We  Thank you for your helpful recommendations.

                                                                                                                                                                                                       we

Reviewer 2 Report

Comments and Suggestions for Authors

Authors investigated the effect of recombinant human bone morphogenetic protein(rh-BMP-2) injection in the gingiva and bone grafting material on the acceleration of the gingival healing process and the proliferation and formation of new epithelial cells in the gingival tissue. The clinical trial includes 20 patients (n= 20) who underwent bilateral implant surgeries in the premolar-molar region of the mandible. Each patient received an implant in both locations, but rh-BMP2 only on the right side. A gingival biopsy was performed 9th day and 4 months after the surgery. The histological result show that the gingival injection of rh-BMP-2 accelerates gingival tissue healing and the proliferation of new epithelial cells in patients undergoing dental implant treatment. The experimental design is reasonable and interesting.

Comments and questions:

1.     The “schema” in line 130 should be capitalized.

2.     Why was the biopsy collected at 9th and 4 months?  

3.     Make sure the ‘9th day’ and ‘9th’, ‘fourth month’ and ‘4 month’ consist of in the manuscript.

4.     The histologically imaging result is not clear, it’s better to be zoomed in.

5.     Delete Figure 4, the information in Table 1 is clear enough.   

6.     Show the main t test results list in Table 3 in Figure 7.  

In summary, I recommend this paper to be minorly revised before accepting.

Comments on the Quality of English Language

Punctuation and capitalization issue.

Author Response

Dear Reviewer,

Reviewer 2

 I hope that your recommendation will help to improve the quality of our manuscript. Thank you.

1-The “schema” in line 130 should be capitalized

Response: Figure 1: Study Design. The schema was capitalized. Moreover, we reduced the text to make it clearer. L148-149.

2-Why was the biopsy collected at “9th” and” 4 months”? 

Response: The biopsy was collected at “9 days” during suture removal because in the literature a complete healing of the epithelium surface was achieved after 10 days in primary intention. A complete healing of the epithelial surface at 9 days confirms a faster healing.  The second biopsy at ” 4 months” is to evaluate the gingival tissue maturation and was performed during the second stage of surgery of the implant.

.

3-Make sure the ‘9th day’ and ‘9th’ , ‘fourth month’ and ‘4 month’ consist of in the manuscript.

Response: Following your request, we added in the introduction this sentence;

The epithelium surface appears to be healed after 10 days” (L93-94) and in the discussion, another sentence “ A complete healing of the epithelial surface at 9 days confirms a faster healing.” (L388-389)

4- The histologically imaging result is not clear, it’s better to be zoomed in.                                  

Response: most of the gingival studies used  the magnification x 40 for histological view  and we adopted this magnification in our research

 We agree with what you demand. For this reason, we framed the area that facilitates us clarity.

5-Delete Figure 4, the information in Table 1 is clear enough

Response: we agree with you. the Figure 4 was removed.

  • Show the main t test results list in Table 3 in Figure 7. 

Response: Excellent idea. We removed Table 3 and adapted the former Figure 7 replaced by

 Figure 3.

Thank you for your helpful recommendations.

Reviewer 3 Report

Comments and Suggestions for Authors

The article is interesting and I want to congratulate the team for performing the study. 

I would like to do some suggestions in order to improve the article:

1. Please give the shape and dimensions of the tested gingival specimen (length, width, thickness)

2. I would suggest to put figure 2 and 3 in one imagine, so it can be easier for the readers to see the differences

3. I would suggest the same for figure 5 and 6

4. I think I will be very interesting to put intraoral pictures at 9 days, in order to see the macroscopic difference in healing (I think clinicians would like to see that)

Author Response

Reviewer 3

I hope that your recommendations will help us to improve the quality of our work.

  • Please give the shape and dimensions of the tested gingival specimen (length, width, thickness)

Response : the shape of the first biopsy  9 day is oval, 2 x 3 mm and the second biopsy at 4 months is circular with a 4 mm of diameter. the thickness in both biopsy is around 2 mm.  These details were added in the manuscript as follow:

9 days after the surgery during suture removal, …. given distally to the implant. The distal part of the implant…..periodontal probe.

  • The biopsy was done using the biopsy forceps from the gingiva 3 mm distally from the implant at the incision level. The oval collected biopsy dimension was 2-3 mm of diameter and around 2 mm of thickness.

And

4 months after the surgery and during the second implant stage …..

periodontal probe.

  • Excision of the tissue covered the head of the implant using a tissue puncher of 4 mm diameter (Cowell Medi Co, Ltd., Seoul, Republic of Korea). The biopsy was circular with a 4 mm of diameter and a thickness of around 2 mm.

  • I would suggest to put figure 2 and 3 in one imagine, so it can be easier for the readers to see the differences

              Response: Excellent idea. Thanks.   The photos were mixed in one new figure (Figure 2).

3- I would suggest the same for figure 5 and 6

Response :we combined the two figures into one. We also combined tables and figures together.

4-. I think I will be very interesting to put intraoral pictures at 9 days, in order to see the macroscopic difference in healing (I think clinicians would like to see that)

Response: At 9 days there was no visible difference at the healed mucosa, For this reason, we didn’t find any utility in adding photos for each group and we opted for the histological gingival healing evaluation (more accurate).

Thank you for your helpful recommendations.

Reviewer 4 Report

Comments and Suggestions for Authors

Congratulation on your work.

Minor comments:

GBR should be explained row 117. Also, please reformulate in order to be clearer where GBR was performed.

The authors should mention the dimension of the biopsy.

Row 126 – reformulate the comment about Ki-27 expression as to belong to the Materials and Methods section.  

The title of figure 4 should be renamed

Please, configurate Table as to be easier to read. Also rename it.

Authors contribution should be elaborated to correspond to the Instructions to authors. Also, rewrite all sections s to be more fit to the journal format.

The references should be updated since the theme is actual and studied in the orofacial area.

Author Response

Reviewer 4

We would like to thank you for your appreciable comments.

the authors should address the following issues to improve the quality of the manuscript:

1-GBR should be explained row 117. Also, please reformulate in order to be clearer where GBR was - performed .

Response : : we followed your request. L 131,132. We added the following sentence:

‘GBR was performed regarding bone dehiscence and exposing of the implant buccally or lingually.”

2-The authors should mention the dimension of the biopsy.

Response : As per your request, we added in the biopsy procedure the sentence colored( in red) L240,241 :

9 days after the surgery during suture removal, …. given distally to the implant. The distal part of the implant…..periodontal probe.

  • The biopsy was done using the biopsy forceps from the gingiva 3 mm distally from the implant at the incision level. The oval collected biopsy dimension was 2-3 mm of diameter and around 2 mm of thickness.

And

4 months after the surgery and during the second implant stage …..

periodontal probe.

Excision of the tissue covered the head of the implant using a tissue puncher of 4 mm diameter (Cowell Medi Co, Ltd., Seoul, Republic of Korea). The biopsy was circular with a 4 mm of diameter and a thickness of around 2 mm. 

3-Row 126 – reformulate the comment about Ki-67 expression as to belong to the Materials and Methods section

Response : as your request we reformulate as following From L143-146.

“The value of ki-67 expression is calculated by counting the cells colored in purple and brown and represents the ratio of new cells and total cells.”

  • The title of Figure 4 should be renamed.

Response : As per your request. we change the title to ” Percentage of complete wound closure in control and test group on “9 days”. But, following another reviewer recommendations, we removed the figure 4 because the table 2 is clear enough.

5-Please, configurate Table as to be easier to read. Also rename it

Response : we agree with you. Thanks.  Table 3 was removed, and an illustration was added to Figure 7 who become Figure 3.

6-Authors contributions should be elaborated to correspond to the Instructions to authors. Also, rewrite all sections to be more fit to the journal format

Response :  Thank you. we adopted the section to the journal format.

7- The references should be updated since the theme is actual and studied in the orofacial area.

Response : : We agree with you. We have updated our references with new ones in the orofacial area. We added the following sentences:

The rh- BMP2 can be used in the Oro-facial area in maxillary sinus augmentation, [4] in alveolar ridge augmentation and preservation, [5,6] in the alveolar defect in cleft palate, [7] in implant and peri-implantitis. [8,9].” (L60-62)

Thank you for your helpful recommendations.

Round 2

Reviewer 1 Report

Comments and Suggestions for Authors

Good look for your future research!

Reviewer 4 Report

Comments and Suggestions for Authors

Thank you for considering my suggestions. 

Good work!